# A *Hidradenitis Suppurativa* molecular disease signature derived from patient samples by high-throughput RNA sequencing and re-analysis of previously reported transcriptomic data sets

**Johannes M. Freudenberg**[1], **Zhi Liu**[2], **Jennifer Singh**[1], **Elizabeth Thomas**[1], **Christopher Traini**[1], **Deepak K. Rajpal**[3], **Christopher J. Sayed**[ID][2]*

**1** GSK Research and Development, Collegeville, PA, United States of America, **2** University of North Carolina Department of Dermatology, Chapel Hill, North Carolina, United States of America, **3** Takeda, Development, Lexington, Massachusetts, United States of America

* csayed@email.unc.edu

**Data Availability Statement:** Most relevant data is available in supplemental tables. Sequencing data

## Abstract

Hidradenitis suppurativa (HS) is a common, debilitating inflammatory skin disease linked to immune dysregulation and abnormalities in follicular structure and function. Several studies have characterized the transcriptomic profile of affected and unaffected skin in small populations. In this study of 20 patients, RNA from lesional and matching non-lesional skin biopsies in 20 subjects were used to identify an expression-based HS disease signature. This was followed by differential expression and pathway enrichment analyses, as well as jointly reanalyzing our findings with previously published transcriptomic profiles. We establish an RNA-Seq based HS expression disease signature that is mostly consistent with previous reports. Bulk-RNA profiles from 104 subjects in 7 previously reported data sets identified a disease signature of 118 differentially regulated genes compared to three control data sets from non-lesional skin. We confirmed previously reported expression profiles and further characterized dysregulation in complement activation and host response to bacteria in disease pathogenesis. Changes in the transcriptome of lesional skin in this cohort of HS patients is consistent with smaller previously reported populations. The findings further support the significance of immune dysregulation, in particular with regard to bacterial response mechanisms. Joint analysis of this and previously reported cohorts indicate a remarkably consistent expression profile.

## Introduction

Hidradenitis suppurativa (HS) is a severe, debilitating chronic inflammatory disease affecting hair follicles with a predilection for intertriginous areas. While historically considered a rare disease, it is clear from recent literature that it relatively common with a prevalence

has been deposited in GEO (Accession number GSE151243).

**Funding:** This study was funded by GlaxoSmithKline. JF, DR, ET and CT were employees of GlaxoSmithKline and had input study design, data collection and analysis, decision to publish, and preparation of the manuscript."

**Competing interests:** Christopher Sayed discloses that he is speaker for Abbvie and Novartis; on advisory boards for Abbvie, UCB and InflaRx; a co-investigator for Abbvie and Novartis; and an investigator for InflaRx, Chemocentryx, Incyte, GlaxoSmithKline and UCB. Johannes Freudenberg, Deepak Rajpal, Elizabeth Thomas, Christopher Traini are current or former employees and shareholders of GlaxoSmithKline. Zhi Liu reports no conflict of interest. This does not alter our adherence to PLOS ONE policies on sharing data and materials.

**Abbreviations:** AQP5, aquaporin 5; DCD, dermcidin; ECM, extracellular matrix; FDR, false discovery rate; HS, hidradenitis suppurativa; TNF, tumor necrosis factor; UNC, University of North Carolina.

of 0.7–1% [1–3] with a predilection for female and African-American patients [4]. Despite its severe impact on quality of life and relatively high frequency in the population, the pathogenesis of HS is still not well understood [5, 6]. There is a pressing need to develop more effective treatment as only one FDA-approved treatment, the TNF-inhibitor adalimumab, currently exists with only 50% of patients achieving a meaningful clinical response [7, 8], especially in later disease when the opportunity to stop progression has been missed [9].

Mechanisms involving follicular occlusion, abnormalities in the skin and follicular microbiome, and dysregulation of inflammatory pathways have been described and supported in relatively small studies and have been recently summarized [6, 10–14]. Furthermore, a number of genetic variants have been reported contributing to HS susceptibility or severity involving genes encoding subunits of the γ-secretase protein complex, connexin-26, fibroblast growth factor, tumor necrosis factor, defensin beta 2 and 3, interleukin-12 receptor, and MyD88 [15]. Consistent with previous reports [13, 14, 16–21], elevated expression was reported for antimicrobial peptides lesional vs non-lesional skin and for inflammatory cytokines including TNF, IL-1, IL-6, IL-10, IL-17, IL-23 and IL-36 in lesional skin and blood of patients with HS [22]. This analysis also discovered increases in complement expression and a plasma cell signature that had not been well-described previously. In addition, dysregulation of keratinocyte function has been implicated [23].

Transcriptome data from a cohort with a total of 17 patients with moderate-to-severe HS and 10 healthy donors has been published [24] and re-analyzed several times [22, 25, 26]. More recently, a number of additional bulk RNA transcriptome datasets in HS cohorts have been published [24, 27–32]. These studies highlighted a number of genes and molecular pathways that are dysregulated in HS skin lesions relative to control (non-lesion or healthy) including immune cell adhesion and complement system [24], antimicrobial peptides [24, 27–32] such as dermcidin and cytokine regulators such as IL-37, interferon pathways and leucocyte activation [24, 27–33], IL17A signalling [24, 27–32], T-cell and B-cell biology [24, 27–32] and the role of apocrine glands [24, 27–32]. In addition, studies with patient samples at single cell resolution report similar findings [24, 27–32] and offer some transcriptomics based insights into potential hypotheses on stem cell infidelity and pathological epidermal remodeling (https://www.biorxiv.org/content/10.1101/2020.04.21.053611v1).

In order to validate previous findings and seek new understanding of disease pathogenesis, we performed RNASeq analysis on lesional and non-lesional skin of 20 subjects with HS and conducted a joint analysis together with the previously published HS transcriptomics studies. The goal of the current study was to generate a high-confidence transcriptome based HS disease signature that is reproducible across HS patient cohorts and platforms, to characterize molecular mechanisms underlying the disease, and to identify molecular pathways that are affected by the disease on the transcriptional level.

## Results

This study enrolled 20 HS patients at the University of North Carolina Department of Dermatology clinics from October 2017 through February 2018. The demographic characteristics of this cohort are shown in Table 1. Whole skin punch biopsies were collected from a lesional site and a matching non-lesional site located 5 cm from the site of inflammation (S1 Table). All samples passed RNA quality assessment thresholds and were sequenced using Illumina paired end sequencing. No mapped transcript bias was observed in the genome-aligned samples and all samples exceeded the minimum recommended depth of 50 million reads per sample. Sequencing data has been deposited in GEO (Accession number GSE151243).

**Table 1. Demographic characteristics of the hidradenitis suppurativa 20 patient cohort.**

| | |
|---|---|
| Average age in years (sd) | |
| Sample collection | 36 (±10.5) |
| At diagnosis | 31.6 (±10.7) |
| At symptom onset | 22.7 (±11.1) |
| Gender (M/F) | 3/17 |
| Race (W/non-W) | 13/7 |
| Smoker (yes/ex/never) | 5/6/9 |
| Average BMI (sd) | 36.4 (±11.6) |
| Family history (y/n) | 8/12 |
| Average IHS4 | 10.2 (±12.7) |
| Treatment at sample collection (n) | |
| No medical treatment | 7 |
| Clindamycin lotion | 4 |
| Clindamycin/rifampin oral | 3 |
| Trimethroprim/Sulfamethoxazole | 1 |
| Spironolactone | 2 |
| Finasteride | 1 |
| Oral contraceptive | 1 |
| Isotretinoin | 1 |
| Site of tissue collection (n) | |
| Axilla | 4 |
| Inguinal | 9 |
| Buttocks | 2 |
| Inframammary | 1 |
| Labia majora | 1 |
| Pubic | 1 |
| Other | 2 |

IHS4: International Hidradenitis Severity Score System

Reads were aligned to the Ensembl human gene model (GRCh38.86) to generate count data which was then further analyzed including differential expression analysis, pathway enrichment analysis, and comparison to an earlier microarray-based HS dataset [24].

## Differential expression analysis

To determine which transcripts were differentially expressed between lesion and non-lesion samples we fit a generalized linear model [34] accounting for the paired design and estimating transcript level variance and fold changes using an empirical Bayes model. The resulting p-values were then adjusted to account for multiple testing [35]. A gene was considered differentially expressed if it was at least three-fold up or down-regulated in the lesion samples and had an adjusted p-value (i.e. False discovery rate, FDR) of 0.1 or less. Using these thresholds, 1,430 protein coding genes were identified as being differentially expressed (Fig 1, S2 Table). Among the top upregulated transcripts are matrix metalloproteinase-1 and 3 (MMP1, 3) which play a role in the breakdown of extracellular matrix proteins, C-X-C motif chemokine ligands 5 and 13 (CXCL5, 13), SERPINB4, and S100 calcium binding proteins A12 and A15 (S100A12, 15). The latter three are thought to be antimicrobial peptides [22] and are well-known to also be

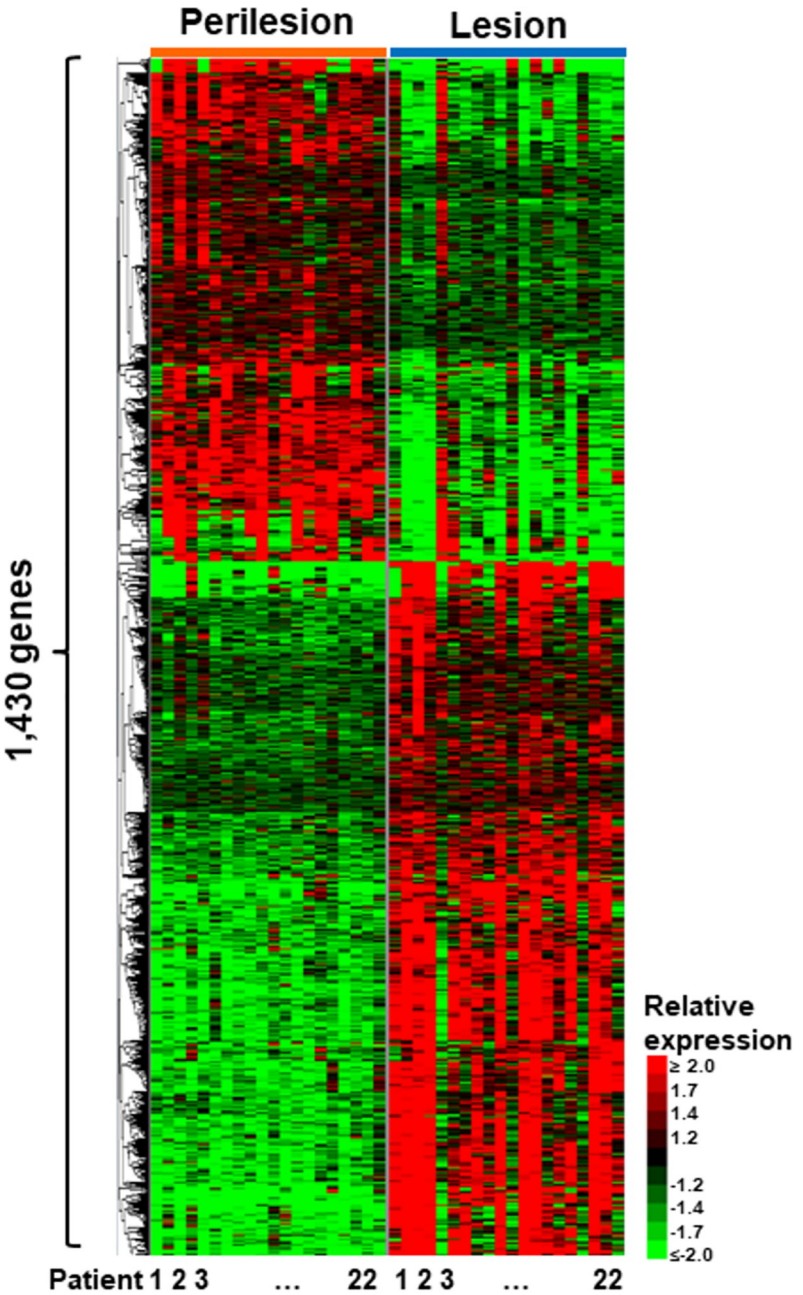

**Fig 1. Overview of the current dataset, differential expression analysis, and comparison to microarray-based study.** 1,430 protein coding genes were identified as being differentially expressed when comparing lesion and non-lesion samples. The heatmap shows the gene expression patterns (rows) for each individual patient sample (columns).

upregulated in psoriasis [36–38] and other inflammatory skin conditions. Interestingly, the antimicrobial peptide dermcidin (DCD) is among the transcripts downregulated in the lesional samples together with prolactin induced protein, secretoglobins, and aquaporin 5 (AQP5) which are associated with sweat gland function [26].

## Pathway enrichment analysis

To explore the underlying molecular mechanisms driving the observed differences in expression, we performed pathway enrichment analysis [39]. Consistent with known disease biology, genes upregulated in lesion samples were involved in both adaptive and innate immune response as well as response to bacterium. Th-17, TNF-alpha, and IL-1beta mediated pathways were upregulated as were neutrophil and lymphocyte activation and ECM remodeling. Keratinization and intermediate filament were among the down-regulated pathways (Fig 2, S3 Table). Several key pathways that were highlighted in a recent review of HS mechanisms [40] were upregulated as well including macrophage response, PI3K/AKT signaling, IL-17 signaling, Jak-STAT signaling, and complement activation (S3 Table).

We then turned our focus on specific pathways known to be involved in HS disease biology (Fig 2). Several important cytokines or their receptors show increased expression in the lesion samples including IL-12, IL-23, IL-17, and IL-36 while IL-1A and IL-18 are decreased. IL-6 and IL-1B which were recently cited [40] were significantly increased (9 fold and 5 fold, respectively). Proteolytic enzymes such as MMP2, MMP9, and MMP12 are all upregulated in the lesion samples while neutrophil elastase (ELANE) expression appears to decrease in these whole skin bulk RNA samples relative to non-lesional skin. Expression measures of antimicrobial peptides such as S100A7 and S100A8 as well as DEFB4A are significantly increased in the lesion samples while expression of the skin-specific DCD which is secreted by the eccrine sweat glands is significantly decreased in the lesion samples (Fig 2A).

Given the recent interest [41, 42], we more closely explored members of the complement pathway (Fig 2). Complement C5a receptor 1 (C5AR1) and complement C3b/C4b receptor 1 (CR1) as well as ficolin 1 (FCN1) were all significantly upregulated in the lesion samples with average fold changes ranging from 3.6 to 5.0. Other pathway members significantly dysregulated (FDR < 0.1) included complement component 5a receptor 2 (C5AR2), CD55, and mannan binding lectin serine peptidase 1 and 2 (MASP1, 2). The complete list of differential expression analysis results can be found in S2 Table.

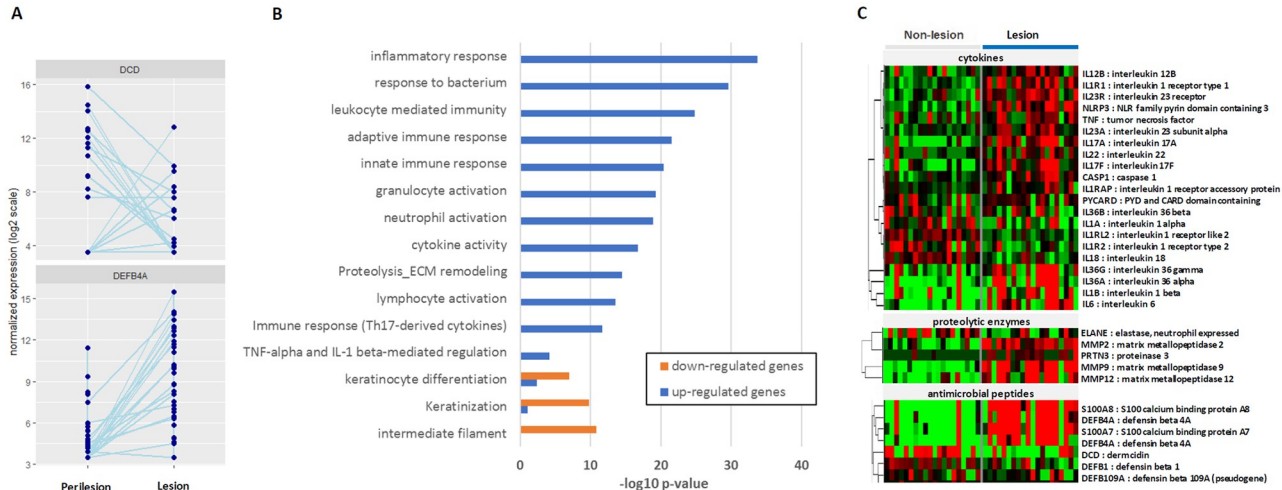

**Fig 2. Pathway enrichment analysis and exploration of disease biology.** Pathway analysis was performed to characterize the underlying molecular mechanisms driving the observed differences in expression. Antimicrobial peptide DEFB4A was significantly increased in the lesion samples while DCD expression was significantly decreased (C). Upregulated pathways include both adaptive and innate immune response as well as response to bacterium, Th-17, TNF-alpha, and IL-1beta mediated pathways, neutrophil and lymphocyte activation and ECM remodeling. Keratinization and intermediate filament were among the down-regulated pathways (B). Key genes that are part of specific pathways known to be involved in HS disease biology are generally upregulated in lesion samples (C).

**Joint reanalysis of HS skin bulk RNA transcriptomics datasets and comparison to the current study.** To compare our results to previously published studies we conducted a reanalysis of seven HS skin bulk RNA datasets [24, 27–32] retrieved from a public repository comparing a total of 104 previous lesion samples to respective controls (71 non-lesional and 43 normal samples) using a vote-counting procedure [43] after conducting differential expression analysis within each study comparing HS lesion samples to their respective controls (see Materials and methods). We chose this approach due to the heterogeneity in study designs and transcriptomic platform technologies (e.g. microarray and RNA-Seq). We identified 118 protein coding genes that were consistently differentially expressed across HS cohorts and platforms in at least eight of the 11 resulting comparisons (including the current study, Fig 3A and S4 Table). All studies assessed bulk RNA expression in either whole skin or apocrine glands comparing HS to peri- or non-lesion from the same patient (paired design) or to similar tissue from healthy volunteers (unpaired design). The consistent patterns across all eight studies suggests overall relatively high concordance between cohorts. Comparing log2 fold change estimates from individual studies for select genes in the HS disease signature showed considerable variation of fold changes and p-values but taken together, this panel is consistently dysregulated in HS (Fig 3B). On the pathway level, the HS disease signature is significantly enriched for leukocyte mediated immunity, inflammatory response, neutrophil degranulation, granulocyte activation, humoral immune response, leukocyte degranulation, host defense, extracellular matrix organization, B cell activation, interleukin-10 signaling, T cell activation, and complement activation. In this skin transcriptomics based signature, we did not observe enrichment for sex hormone related pathways such as estrogen signaling or androgen metabolism (Table 2).

## Discussion

In many ways, the results from the newly analyzed profiles confirm and build upon previously reported findings that demonstrate a key role for immune dysregulation as a component of HS pathogenesis. Similarly, expression of genes associated with extracellular matrix protein breakdown and wound repair such as MMP1 and MMP3 is prominent [22, 26, 44]. These findings should not be surprising given the cycle of tissue destruction repair that results from HS lesions.

Downregulation of DCD and sweat gland-associated transcripts such as prolactin induced protein, secretoglobins, and AQP5 in lesional vs. non-lesional skin are also consistent with previous studies [26]. While the role of apocrine and eccrine sweat glands in HS has been the subject of much debate, it is difficult to interpret these findings. Given the intense tissue inflammation and destruction in lesional skin the reduced expression of these genes may reflect lack of expression in the general setting of inflammation and new tissue repair, but, in contrast to lesional HS skin, wounded skin has been shown to have increased DCD and AQP5 expression [26]. Notably, sweat glands have been shown to have a potential role in wound repair [45, 46] and it is possible that altered sweat gland structure and function contributes to impaired wound healing in HS. Since HS can appear in locations not typically associated with apocrine glands such as the flank and posterior neck the pathogenesis is unlikely to rest with apocrine glands alone.

Prominent upregulation of pathways known to be associated with HS including Th-17, TNF-α, and IL-1β are once again features common with many previous reports [47–52]. Further, increases in classic complement protein expression, particularly those that are cleaved to form anaphylatoxins such a C4 and C5 are consistent with previous reports. We additionally report increased activation of the mannose binding lectin pathway with upregulated ficolin, mannose binding lectin 2, and MASP1 expression. Conversely, CD55, which acts as a check

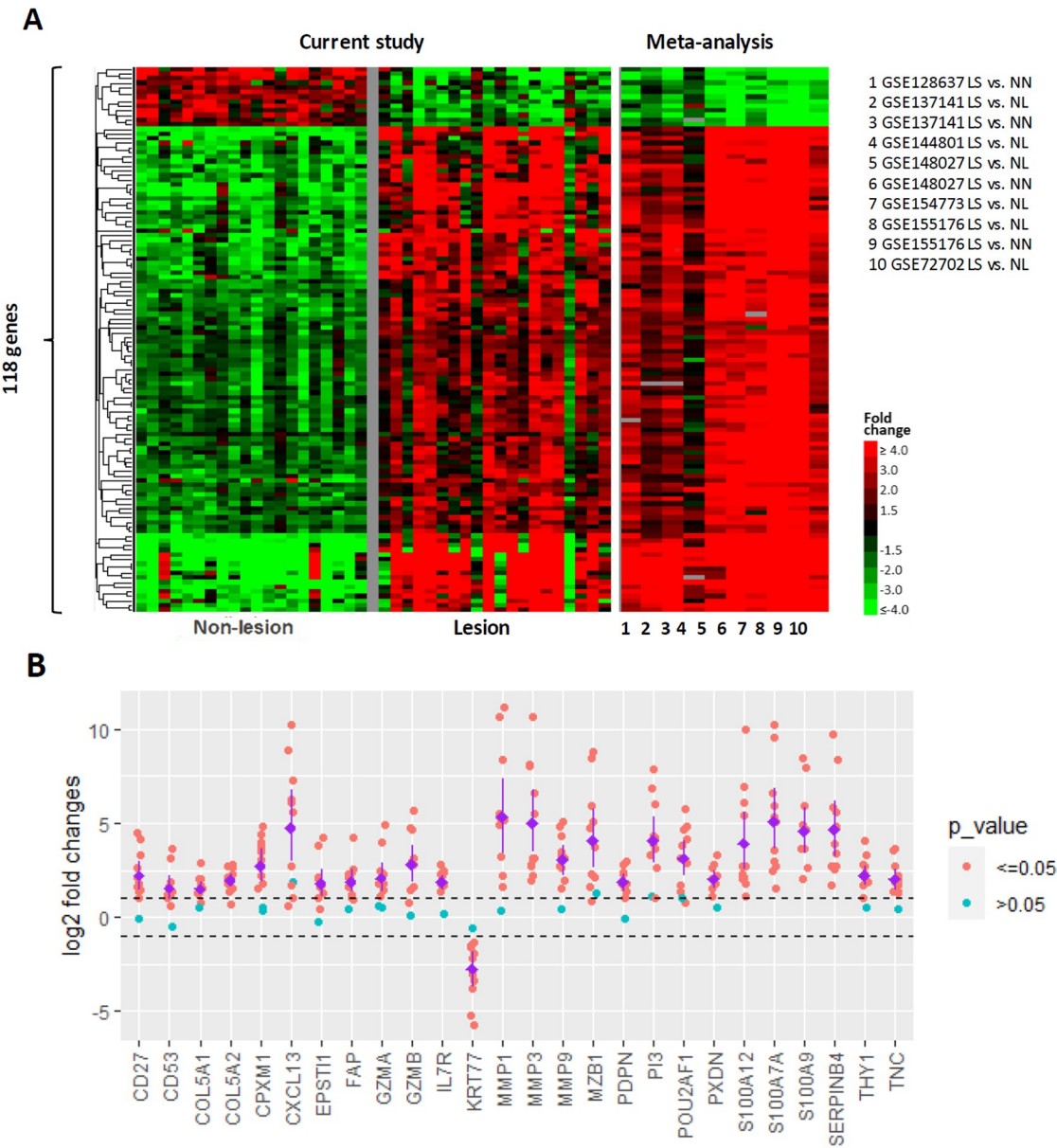

**Fig 3. Joint re-analysis of HS skin bulk RNA transcriptomics datasets and comparison to the current study.** 118 protein coding genes were identified as being consistently differentially expressed when comparing lesion and non-lesion samples across HS cohorts and platforms (A). The heatmap shows the gene expression patterns (rows) for each individual patient sample in the current study and average fold changes for the previously published datasets (columns). The consistent patterns across studies suggests overall relatively high concordance between cohorts. Log2 fold change estimates in the current and previously published studies for select genes in the HS disease signature are shown for individual cohorts (B). The dashed lines represent two fold differences in either direction, blue and red dots represent significance levels, and purple dots and lines represent range and average fold changes. Even though fold changes and p-values vary considerably, it is clear that this panel is consistently dysregulated in HS.

on complement-mediated inflammation by binding to and breaking down complement proteins, is downregulated in lesional HS. This indicates further dysregulation and over activity of complement-mediated inflammation. Inhibitors of C5a have shown early promising results in the treatment of HS, and it is possible that other components of this cascade have therapeutic relevance [41, 53].

**Table 2. Molecular pathways enriched in HS disease signature derived by re-analysis of existing transcriptomic data sets.**

| Pathway | Genes from HS signature |
|---|---|
| *leukocyte mediated immunity* | C1QB, CD14, CD19, CD27, CD53, COTL1, CXCL1, CYBB, FCN1, GZMB, IL7R, ITGB2, LILRB2, LILRB3, LTF, LYZ, MMP9, PLAC8, PTPRC, RAB31, S100A12, S100A8, S100A9, SASH3, SELL, SERPINB3, SERPINB4, SLAMF7, TCN1, TNFAIP6 |
| *inflammatory response* | AIM2, C1QB, CCR1, CCR5, CCR7, CD14, CD19, CFB, CXCL1, CXCL10, CXCL13, CXCR4, CYBB, GBP5, IDO1, ITGB2, KRT16, LY86, LYZ, MMP3, MMP9, PTPRC, S100A12, S100A8, S100A9, SPP1, TNFAIP6 |
| *neutrophil degranulation* | CD14, CD53, COTL1, CXCL1, CYBB, FCN1, ITGB2, LILRB2, LILRB3, LTF, LYZ, MMP9, PLAC8, PTPRC, RAB31, S100A12, S100A8, S100A9, SELL, SERPINB3, TCN1, TNFAIP6 |
| *granulocyte activation* | CD14, CD53, COTL1, CXCL1, CYBB, FCN1, ITGB2, LILRB2, LILRB3, LTF, LYZ, MMP9, PLAC8, PTPRC, RAB31, S100A12, S100A8, S100A9, SELL, SERPINB3, TCN1, TNFAIP6 |
| *humoral immune response* | C1QB, CCR7, CD19, CFB, CXCL1, CXCL10, CXCL13, FCN1, LTF, LYZ, PI3, POU2AF1, PTPRC, S100A12, S100A8, S100A9 |
| *leukocyte degranulation* | CD14, CD53, COTL1, CXCL1, CYBB, FCN1, ITGB2, LILRB2, LILRB3, LTF, LYZ, MMP9, PLAC8, PTPRC, RAB31, S100A12, S100A8, S100A9, SELL, SERPINB3, TCN1, TNFAIP6 |
| *response to other organism* | AIM2, CCR5, CCR7, CD14, COCH, COTL1, CXCL1, CXCL10, CXCL13, CXCR4, GBP5, GZMA, IDO1, IL10RA, LILRB2, LTF, LY86, LYZ, PI3, PIM2, PLAC8, PTPRC, S100A12, S100A8, S100A9 |
| *Extracellular matrix organization* | ADAM12, COL4A1, COL5A1, COL5A2, FBN2, ITGB2, MMP1, MMP3, MMP9, PXDN, SERPINH1, SPP1, TNC |
| *B cell activation* | CD19, CD27, CD38, CD79A, IL7R, MZB1, PTPRC, SASH3, TNFSF13B |
| *Interleukin-10 signaling* | CCR1, CCR5, CXCL1, CXCL10, IL10RA |
| *T cell activation* | CCR7, CD27, CD3D, EOMES, IDO1, IL7R, LILRB2, PTPRC, RHOH, SASH3, THY1, TNFSF13B |
| complement activation | C1QB, CD19, CFB, FCN1 |

One of the most striking associations identified in the current dataset is with bacterial response mechanisms, which is reflected in the combination of upregulated AMPs, complement signaling, and inflammatory pathways associated with Th-17, TNF-α, and IL-1β. The clinical features of HS such as abscesses, erythema, warmth, pustules, and purulent drainage are identical in many ways to the presentation of acute bacterial infections such bacterial abscesses, cellulitis, and folliculitis. HS is set apart from cutaneous infections clinically as it is not contagious and leads to chronic/recurrent inflammation that does not fully respond to antibiotics. HS does, however, improve temporarily with broad spectrum antibiotics or antibiotic regimens tailored to the patient microbiome [54–56]. This indicates that commensal bacteria likely play a role in activating the immune response that mimics the response seen to more classic pathogens such as Staphylococcus aureus, which is identified infrequently in HS lesions. This is possibly related to complement activation by bacteria and subsequent production of anaphylatoxins, though other players in the innate immune response may also be involved [57, 58]. Characteristic microbiome patterns in patients with HS have been established [59–61], but it is unclear if this dysbiosis is a result of abnormal host defenses, an abnormality in follicular structure or function, a combination of these factors, or perhaps merely a byproduct of the inflammatory changes associated of HS. Whether this dysbiosis plays a causal role triggering follicular inflammation or the inflammatory environment of HS leads to characteristic changes in the microbiome is unclear, but the role of commensal bacteria in amplifying and sustaining inflammation in the disease seems certain.

Joint re-analysis of previously reported data sets in conjunction with these newly reported profiles demonstrates a strikingly consistent disease signature of 118 dysregulated genes in HS lesional skin (Table 1). Consistent with our new cohort, the profile indicates that upregulated neutrophil and leukocyte mediated immunity, complement activation, response to bacteria and extracellular matrix remodeling are the predominant pathways activated in HS lesions. While the re-analysis allows for a larger sample size than individual previous reports, it is limited in that it reflects a predominantly Caucasian population from North America and Western Europe. Other challenges include heterogeneity due to varying study designs and transcriptomic platform technologies used. We employed a simple vote-counting method to jointly analyze the current and prior data to sidestep these challenges. A more complex formal meta-analysis could potentially address cohort heterogeneity but integrating microarray and RNA-Seq data in such analysis remains challenging. Our own dataset is also limited since most patients had moderate or severe disease that required surgical intervention, so these findings may not be generalizable to earlier stages of disease. All datasets examined are also limited by a lack of characterization of the histological findings in the tissue, such as the presence of epithelialized tunnels, which likely have an impact on expression patterns and contribute to heterogeneity of the findings [62].

Future studies should aim to characterize more diverse and well-characterized populations with more information on disease severity, course, and comorbidities. They should also strive to include patients before and after starting a variety of treatments in significant numbers to help understand how expression profiles may be impacted with various interventions and predict treatment response. Further studies utilizing single-cell and spatial transcriptomic analysis also have the potential to better characterize HS pathophysiology.

## Materials and methods

### Subjects

Patients were recruited in a subspecialty hidradenitis suppurativa clinic at the University of North Carolina (UNC) at Chapel Hill Department of Dermatology from October 2017 through February 2018. All study protocols were approved by the UNC Institutional Review board. All adult patients provided written informed consent and for minors assent was obtained from the subject along with consent from legal guardians prior to enrollment. Subjects age 13 years and older with a clinical diagnosis of HS based on findings of 1) abscess, nodules and/or cutaneous sinuses 2) in classic intertriginous locations 3) for greater than 6 months were included. Recruitment was limited to patients already undergoing medically necessary surgical procedures in which tissue would be collected as part of clinical care. Subjects taking medications targeting TNFα, IL-1, IL-17, IL-23 or immunomodulators that may otherwise significantly alter inflammatory profiles at the time of the specimen collection or within 30 days prior were excluded.

### Specimen collection

Immediately prior to planned surgical procedures, four millimeter punch biopsies were obtained from both lesional and non-lesional skin. Non-lesional biopsies were performed approximately 5 centimeters from any signs of cutaneous inflammation in the same body region as the lesional biopsy. Specimens were immediately placed in RNALater© (Ambion Inc, Foster City, CA) for storage and transport to the research laboratory.

### RNA extraction

Punch biopsies from lesional and non-lesional skin areas were transferred to 1.5 ml RNAlater tissue protect tubes (Qiagen, Germantown, MD). Samples were kept at 4 ˚C for 24 h after

which they were transferred and stored at − 80 ˚C. Biopsy samples were lysed using a mechanical bead-based disruption method. RNA was extracted from the minced tissue using the Qiagen RNeasy Mini Kit following manufacturer's recommendations for fibrous tissue RNA isolation (Qiagen, Germantown, MD). The RNA was stored at − 80 ˚C until further use.

## RNA-seq methods

**Data generation.** RNA from lesional and non-lesional cohorts, consisting of twenty patients each, were used for RNA-seq analyses. A total of 350 nanograms of total RNA was rRNA depleted using the Ribo-Zero rRNA Removal Kit (Epicentre, Madison, WI, USA). Sequencing libraries were generated using the Illumina Truseq Total stranded kits (Illumina, San Diego, CA) using 12 PCR cycles for amplification. Library quality control was performed on the 2200 TapeStation Bioanalyzer (Agilent Technologies, Santa Clara, CA, USA) and quantified using the KAPA library quantification kit (Kapa Biosystems, Wilmington, MA, USA). RNA-seq was performed on the HiSeq 3000 system (Illumina, San Diego, CA, USA) using the 100-cycle paired-end sequencing protocol with five samples pooled in one lane.

**Preprocessing.** We used Skewer v0.2.218 to trim RNA-seq reads with end quality of less than 20. Additionally, we also removed reads with an average read quality of less than 20 or reads that were trimmed to less than 30 nucleotides. We aligned RNA-seq reads to the human reference genome (Genome Reference Consortium Human Build 38 assembly) using "Splice Transcripts Alignment to a Reference" tool (STAR, version 2.5.1b). We then used featurecounts tool from the Subread package v1.4.6-p5 to generate count data with the following parameters "-s 2—primary -p—C". Ensembl human gene model (GRCh38.86) was used to generate count data. We used the edgeR v 3.421 and limma v 3.422 package to compute counts per million (CPM) values for each transcript and sample analysis as follows. First, we removed transcripts that were considered not expressed, i.e. where CPM ≤ 0.5 in more than 80% of the samples. We then normalized the samples using the calcNormFactors() function and finally, we log2 transformed the resulting normalized CPM values for subsequent analyses.

**Differential expression, cluster, and enrichment analysis.** To determine differential mRNA expression, a linear model was fit [34] on the transcript count data. The false discovery rate (FDR) was computed as an adjusted p-value to account for multiple testing [35] and a cut-off of 10% FDR together with an absolute fold change cutoff of 3.0 or more was used to define differential expression. For cluster analysis and to functionally annotate sets of differentially regulated genes we used a pathway enrichment analysis tool based on the hypergeometric test [39]. Gene pathway annotations (GO, KEGG, Reatome) were retrieved from Bioconductor [63]. For comparison, published HS datasets [24, 27–32] were reanalyzed. Raw data files (CEL files) were downloaded from the NCBI Gene Expression Omnibus (GEO) [64] for Affymetrix platforms where available and pre-processed using the RMA (Robust Multi-chip Average) pipeline [65] in combination with the must current re-annotated probeset definitions [66] (Version 21). For non-Affymetrix platforms, the already pre-processed data was used. To determine differential mRNA expression for each study, a linear model was fit [67, 68] for microarray data and for RNA-Seq data [34], respectively. The false discovery rate (FDR) was computed as an adjusted p-value [35] to account for multiple testing and a cut-off of 10% FDR was used to define differential expression. We then used a vote-counting procedure [43] to determine a consensus HS signature by selecting all genes that were found significantly differentially expressed in 8 out of 10 comparisons.

## Supporting information

**S1 Table. Subject demographics and clinical characteristics.**
(XLSX)

**S2 Table. Full differential gene expression data.**
(XLSX)

**S3 Table. Full pathway enrichment analysis.**
(XLSX)

**S4 Table. Differential gene expression among jointly analyzed cohorts.**
(XLSX)

## Acknowledgments

The authors would like to acknowledge David Rubenstein, who was instrumental in establishing this collaboration and Akanksha Gupta for providing early project coordination.

## Author Contributions

**Conceptualization:** Johannes M. Freudenberg, Zhi Liu, Deepak K. Rajpal, Christopher J. Sayed.

**Data curation:** Johannes M. Freudenberg, Christopher J. Sayed.

**Formal analysis:** Johannes M. Freudenberg, Christopher J. Sayed.

**Funding acquisition:** Christopher J. Sayed.

**Investigation:** Johannes M. Freudenberg, Jennifer Singh, Elizabeth Thomas, Christopher Traini, Deepak K. Rajpal, Christopher J. Sayed.

**Methodology:** Johannes M. Freudenberg, Deepak K. Rajpal, Christopher J. Sayed.

**Project administration:** Elizabeth Thomas, Christopher J. Sayed.

**Resources:** Christopher J. Sayed.

**Supervision:** Zhi Liu, Elizabeth Thomas, Deepak K. Rajpal, Christopher J. Sayed.

**Validation:** Johannes M. Freudenberg.

**Visualization:** Johannes M. Freudenberg.

**Writing – original draft:** Johannes M. Freudenberg, Christopher J. Sayed.

**Writing – review & editing:** Johannes M. Freudenberg, Zhi Liu, Jennifer Singh, Elizabeth Thomas, Christopher Traini, Deepak K. Rajpal, Christopher J. Sayed.

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
