## [Decision Letter · Decision Letter 0]

19 Oct 2022

PONE-D-22-15397A hidradenitis suppurativa molecular disease signature derived from patient samples by high-throughput RNA sequencing and meta-analysis of previously reported transcriptomic data setsPLOS ONE

Dear Dr. Sayed,

Thank you for submitting your manuscript to PLOS ONE. After careful consideration, we feel that it has merit but does not fully meet PLOS ONE’s publication criteria as it currently stands. Therefore, we invite you to submit a revised version of the manuscript that addresses the points raised during the review process.

I would like to sincerely apologise for the delay you have incurred with your submission. We have now received five completed reviews; the comments are available below. The reviewers have raised significant scientific concerns about the study that need to be addressed in a revision.

Please revise the manuscript to address all the reviewer's comments in a point-by-point response in order to ensure it is meeting the journal's publication criteria. Please note that the revised manuscript will need to undergo further review, we thus cannot at this point anticipate the outcome of the evaluation process.

We look forward to receiving your revised manuscript.

Kind regards,

Miquel Vall-llosera Camps

Senior Editor

PLOS ONE

Journal Requirements:

Christopher Sayed discloses that he is speaker for Abbvie and Novartis; on advisory boards for Abbvie, UCB and InflaRx; a co-investigator for Abbvie and Novartis; and an investigator for InflaRx, Chemocentryx, Incyte, GlaxoSmithKline and UCB.

Johannes Freudenberg, Deepak Rajpal, Elizabeth Thomas, Christopher Traini are current or former employees and shareholders of GlaxoSmithKline.

Zhi Liu reports no conflict of interest.

Reviewers' comments:

Reviewer's Responses to Questions

**Comments to the Author**

1. Is the manuscript technically sound, and do the data support the conclusions?

Reviewer #1: Yes

Reviewer #2: Yes

Reviewer #3: Yes

Reviewer #4: Partly

Reviewer #5: Yes

2. Has the statistical analysis been performed appropriately and rigorously? 

Reviewer #1: Yes

Reviewer #2: Yes

Reviewer #3: Yes

Reviewer #4: No

Reviewer #5: Yes

3. Have the authors made all data underlying the findings in their manuscript fully available?

Reviewer #1: Yes

Reviewer #2: Yes

Reviewer #3: Yes

Reviewer #4: Yes

Reviewer #5: Yes

4. Is the manuscript presented in an intelligible fashion and written in standard English?

Reviewer #1: Yes

Reviewer #2: Yes

Reviewer #3: Yes

Reviewer #4: Yes

Reviewer #5: Yes

5. Review Comments to the Author

Reviewer #1: This is a very interesting work about the molecular disease signature derived from HS patient samples by high-throughput RNA sequencing and meta-analysis of previously reported transcriptomic data sets. This work is well written and organized.

However, please consider these points:

1. Page 3, line 48. Please consider this work about HS and Adalimumab mainly focusing on the concept of “window of opportunity”: Marzano AV, et al. Evidence for a 'window of opportunity' in hidradenitis suppurativa treated with adalimumab: a retrospective, real-life multicentre cohort study. Br J Dermatol. 2021;184(1):133-140. doi: 10.1111/bjd.18983.

2. Page 3, from line 51. Please also consider the involvement of keratinization and autoinflammatory genes such as GJB2, NLRP3 et al. We invite you to consider these works and explore the concept of the polygenic autoinflammatory nature of HS:

a) Moltrasio C, et al. Hidradenitis Suppurativa: A Perspective on Genetic Factors Involved in the Disease. Biomedicines. 2022;10(8):2039. doi: 10.3390/biomedicines10082039.

b) Marzano AV, et al. Whole-Exome Sequencing in 10 Unrelated Patients with Syndromic Hidradenitis Suppurativa: A Preliminary Step for a Genotype-Phenotype Correlation. Dermatology. 2022;238(5):860-869. doi: 10.1159/000521263.

3. The table 1 should be expanded. Add other clinical items such as age at diagnosis, age at onset, comorbidities…

Reviewer #2: This paper describes an RNASeq analysis on lesional and non-lesional skin of 20 subjects with HS and a meta-analysis together with the previously published HS transcriptomics studies.

The authors used a routine method and analyses to strengthen the findings on hidradenitis suppurativa biological pathways involved in skin inflammation.

This study lacks the clinical characterization of the patients analyzed

Major Comments

1. The clinical characteristics of the subjects are poorly described. We could imagine the cohort the Authors studied is enriched for severe patients undergoing a surgical procedure to alleviate HS. It is not clear either how these patients were treated before the surgical procedure as these treatments could impact biological pathways found by the RNA-seq analysis.

2. It is not clear to me how the non-lesional area was chosen. It is known that dermal lesions and fistula can be present in the dermis without apparent clinical signs on the skin surface. This could have severely affected the genes and pathways found by the RNA-seq experiment.

Minor comments:

1. Introduction- lines 51 to 54 - not all the genes reported in this paragraph have been associated with HS susceptibility. MyD88 and IL-12r, for instance, have been associated only with HS severity. Please correct this part.

2. Introduction- lines 54 to 56 - where the measures for these cytokines were done is unclear. Blood? Skin? Lesional vs non-lesional skin? HS patients vs controls? This should be better explained

Reviewer #3: Thankyou for the opportunity to review this manuscript- A highly novel and insightful piece with high standard statistcal analyses.

The only discussion point I would add is the inherent limitation in the selection of biopsy site in previous studies. The presence of epithelialised tunnels is known to impact the gene expression and protein expression in HS samples (Navrazhina et al JACI 2021), and this should be acknowledged as a potential source of heterogeneity.

Reviewer #4: The manuscript by Johannes Freudenberg et al. describes the transcriptomic signature of skin samples (lesional and perilesional) from 20 patients with HS. The manuscript is well written, the RNA-Seq is appropriate, and the results and discussion are consistent with the analysis reported. A meta-analysis of publicly available transcriptome datasets was also performed, confirming their findings.

(1) The authors are off to a good start, however, the methodology fails to address how the meta-analysis was performed. The authors should include more information that clarifies and justifies their choice of methods, especially regarding the heterogeneity of the studies. Since the meta-analysis is performed using different datasets (high throughput sequencing that quantifies mRNA short read counts and microarray that analyzes the relative amount of mRNA with a fluorescence scanner) and different controls (HS non-lesional skin and healthy control skin, both of them having different molecular patterns), the authors should provide heterogeneity estimates and demonstrate how they dealt with them. Heterogeneity could be one of the reasons for log2 fold change variance cited at page 7, line 147, "Comparing log2 fold change estimates from individual studies for select genes in the HS disease signature showed considerable variation of fold changes and p-values but taken together, this panel is consistently dysregulated in HS (Figure 3B)". If not possible, avoiding high heterogeneity or changing the analysis method would provide more reliable results.

(2) I advise the authors to clarify any study heterogeneity as a limitation alongside the others already reported at line 212, page 10.

(3) In line 145, page 7, the authors state: "We identified 118 protein coding genes that were consistently differentially expressed across HS cohorts and platforms in at least eight of the 11 resulting comparisons". Since the studies were heterogeneous, I suggest the authors highlight the condition of these comparisons, explaining the type of control and array used by them.

(4) Paraphrasing the authors "several studies have characterized the transcriptomic profile of affected and unaffected skin in small populations" (abstract). The authors reported interesting findings regarding complement activation and bacterial response. Thus, if possible, it would be nice for the authors to provide histopathological data such as immunostaining or immunoblot or even western blot showing these markers variation at the translational level. These correlative analyses would increase the impact of the current manuscript and differentiate it from previous works.

(5) Finally, it appears to me that figure 1 is incomplete. To avoid confusion, the authors should include the images corresponding to the letter B and C into the figure 1 as described on its caption or remove B and C from it.

Reviewer #5: This is a well executed study and analysis. My only minor comment is that on Page 9, lines 196-197 data does not quite support the assertion that complement pathways play a driving role. There is dysregulation of the immune response to commensal bacteria, but other pathways may also be involved. This should be edited for clarity.

6. PLOS authors have the option to publish the peer review history of their article (what does this mean?). If published, this will include your full peer review and any attached files.

Reviewer #1: **Yes: **Chiara Moltrasio

Reviewer #2: No

Reviewer #3: No

Reviewer #4: No

Reviewer #5: No

---

## [Author Response · Author response to Decision Letter 0]

5 Jan 2023

Reviewer #1: This is a very interesting work about the molecular disease signature derived from HS patient samples by high-throughput RNA sequencing and meta-analysis of previously reported transcriptomic data sets. This work is well written and organized.

However, please consider these points:

1. Page 3, line 48. Please consider this work about HS and Adalimumab mainly focusing on the concept of “window of opportunity”: Marzano AV, et al. Evidence for a 'window of opportunity' in hidradenitis suppurativa treated with adalimumab: a retrospective, real-life multicentre cohort study. Br J Dermatol. 2021;184(1):133-140. doi: 10.1111/bjd.18983.

Commentary and reference has been added.

2. Page 3, from line 51. Please also consider the involvement of keratinization and autoinflammatory genes such as GJB2, NLRP3 et al. We invite you to consider these works and explore the concept of the polygenic autoinflammatory nature of HS:

a) Moltrasio C, et al. Hidradenitis Suppurativa: A Perspective on Genetic Factors Involved in the Disease. Biomedicines. 2022;10(8):2039. doi: 10.3390/biomedicines10082039.

b) Marzano AV, et al. Whole-Exome Sequencing in 10 Unrelated Patients with Syndromic Hidradenitis Suppurativa: A Preliminary Step for a Genotype-Phenotype Correlation. Dermatology. 2022;238(5):860-869. doi: 10.1159/000521263.

We have included these references and added these genes to the manuscript.

3. The table 1 should be expanded. Add other clinical items such as age at diagnosis, age at onset, comorbidities…

Comorbidities were not included, but age at diagnosis, symptom onset, mean IHS4, and concomitant medication use are now included in table 1

Reviewer #2: This paper describes an RNASeq analysis on lesional and non-lesional skin of 20 subjects with HS and a meta-analysis together with the previously published HS transcriptomics studies.

The authors used a routine method and analyses to strengthen thTe findings on hidradenitis suppurativa biological pathways involved in skin inflammation.

This study lacks the clinical characterization of the patients analyzed

Major Comments

1. The clinical characteristics of the subjects are poorly described. We could imagine the cohort the Authors studied is enriched for severe patients undergoing a surgical procedure to alleviate HS. It is not clear either how these patients were treated before the surgical procedure as these treatments could impact biological pathways found by the RNA-seq analysis.

We have expanded table one to describe additional clinical characteristics including IHS4 score as a measure of severity and concomitant medications. Those on biologics at the time of tissue collection were intentionally excluded to minimize impact of immunomodulation on expression patterns. Full clinical data will be made publicly available alongside the transcriptome data following publication.

2. It is not clear to me how the non-lesional area was chosen. It is known that dermal lesions and fistula can be present in the dermis without apparent clinical signs on the skin surface. This could have severely affected the genes and pathways found by the RNA-seq experiment.

In each of these cases, all involved tissue was carefully examined and probed intraoperatively to delineate the margins of clinically involved tissue. Any non-lesional skin was at least 5 centimeters away and had no apparent inflammatory changes are scarring present. While it is impossible to be certain, we feel highly confident that this skin was not significantly inflamed and acts as a reasonable comparative control. Going further away than this would risk losing site-specific background features of the body region, which would lead to more confusion in interpretation. Comparing skin of the axillae to the lateral trunk or upper arm without the typical follicles or adnexal structures seen in the involved area could lead to many differences in expression that are merely site specific.

Minor comments:

1. Introduction- lines 51 to 54 - not all the genes reported in this paragraph have been associated with HS susceptibility. MyD88 and IL-12r, for instance, have been associated only with HS severity. Please correct this part.

This is amended to point out that some are related to susceptibility and some to severity

2. Introduction- lines 54 to 56 - where the measures for these cytokines were done is unclear. Blood? Skin? Lesional vs non-lesional skin? HS patients vs controls? This should be better explained

This has been added

Reviewer #3: Thankyou for the opportunity to review this manuscript- A highly novel and insightful piece with high standard statistcal analyses.

The only discussion point I would add is the inherent limitation in the selection of biopsy site in previous studies. The presence of epithelialised tunnels is known to impact the gene expression and protein expression in HS samples (Navrazhina et al JACI 2021), and this should be acknowledged as a potential source of heterogeneity.

We have added this commentary in discussion of limitations and included the reference.

Reviewer #4: The manuscript by Johannes Freudenberg et al. describes the transcriptomic signature of skin samples (lesional and perilesional) from 20 patients with HS. The manuscript is well written, the RNA-Seq is appropriate, and the results and discussion are consistent with the analysis reported. A meta-analysis of publicly available transcriptome datasets was also performed, confirming their findings.

(1) The authors are off to a good start, however, the methodology fails to address how the meta-analysis was performed. The authors should include more information that clarifies and justifies their choice of methods, especially regarding the heterogeneity of the studies. Since the meta-analysis is performed using different datasets (high throughput sequencing that quantifies mRNA short read counts and microarray that analyzes the relative amount of mRNA with a fluorescence scanner) and different controls (HS non-lesional skin and healthy control skin, both of them having different molecular patterns), the authors should provide heterogeneity estimates and demonstrate how they dealt with them. Heterogeneity could be one of the reasons for log2 fold change variance cited at page 7, line 147, "Comparing log2 fold change estimates from individual studies for select genes in the HS disease signature showed considerable variation of fold changes and p-values but taken together, this panel is consistently dysregulated in HS (Figure 3B)". If not possible, avoiding high heterogeneity or changing the analysis method would provide more reliable results.

The reviewer raises a great point. We are using a vote-counting procedure [Bushman & Wang] after conducting differential expression analysis within each study individually. We chose this method exactly for the reasons the reviewer pointed out. We have added more detail to the text to clarify the method used.

(2) I advise the authors to clarify any study heterogeneity as a limitation alongside the others already reported at line 212, page 10.

Thank you for flagging this. We have added a sentence regarding study heterogeneity to the section as suggested by the reviewer.

(3) In line 145, page 7, the authors state: "We identified 118 protein coding genes that were consistently differentially expressed across HS cohorts and platforms in at least eight of the 11 resulting comparisons". Since the studies were heterogeneous, I suggest the authors highlight the condition of these comparisons, explaining the type of control and array used by them.

This is again a great point. We have added a statement describing the controls and transcriptomic platforms used.



(4) Paraphrasing the authors "several studies have characterized the transcriptomic profile of affected and unaffected skin in small populations" (abstract). The authors reported interesting findings regarding complement activation and bacterial response. Thus, if possible, it would be nice for the authors to provide histopathological data such as immunostaining or immunoblot or even western blot showing these markers variation at the translational level. These correlative analyses would increase the impact of the current manuscript and differentiate it from previous works.

We agree that this would be and interesting direction for study. We currently have some of this work evaluation complement activation and bacterial response in our labs, but it is currently not ready for publication and likely will be part of a larger and separate project that expands on this work.

(5) Finally, it appears to me that figure 1 is incomplete. To avoid confusion, the authors should include the images corresponding to the letter B and C into the figure 1 as described on its caption or remove B and C from it.

Thank you for catching this. We had updated our figures and somehow did not remove the extra information for the caption. Reference to panels B and C have been removed.

Reviewer #5: This is a well executed study and analysis. My only minor comment is that on Page 9, lines 196-197 data does not quite support the assertion that complement pathways play a driving role. There is dysregulation of the immune response to commensal bacteria, but other pathways may also be involved. This should be edited for clarity.

This suggestion has been softened with mention that other aspects of the innate immune response may also be players. References have been added for support.

---

## [Decision Letter · Decision Letter 1]

2 Feb 2023

PONE-D-22-15397R1A hidradenitis suppurativa molecular disease signature derived from patient samples by high-throughput RNA sequencing and meta-analysis of previously reported transcriptomic data setsPLOS ONE

Dear Dr. Sayed,

Thank you for submitting your manuscript to PLOS ONE. After careful consideration, we feel that it has merit but does not fully meet PLOS ONE’s publication criteria as it currently stands. Therefore, we invite you to submit a revised version of the manuscript that addresses the points raised during the review process.

Please carefully answer the technical points raised by Reviewer 4, in particular I recommend to perform the type of analysis suggested. Concerning Reviewer 1, please correct the typo line 49, and delete *GJB2* line 55 since it is the name of connexin-26 gene. Since, if I correctly understand, you mentionned the name of proteins it is unnecessary to italicize.

We look forward to receiving your revised manuscript.

Kind regards,

Michel Simon, Ph. D.

Academic Editor

PLOS ONE

Journal Requirements:

Reviewers' comments:

Reviewer's Responses to Questions

**Comments to the Author**

1. If the authors have adequately addressed your comments raised in a previous round of review and you feel that this manuscript is now acceptable for publication, you may indicate that here to bypass the “Comments to the Author” section, enter your conflict of interest statement in the “Confidential to Editor” section, and submit your "Accept" recommendation.

Reviewer #1: All comments have been addressed

Reviewer #3: All comments have been addressed

Reviewer #4: (No Response)

2. Is the manuscript technically sound, and do the data support the conclusions?

Reviewer #1: Yes

Reviewer #3: Yes

Reviewer #4: Partly

3. Has the statistical analysis been performed appropriately and rigorously? 

Reviewer #1: Yes

Reviewer #3: Yes

Reviewer #4: No

4. Have the authors made all data underlying the findings in their manuscript fully available?

Reviewer #1: Yes

Reviewer #3: Yes

Reviewer #4: Yes

5. Is the manuscript presented in an intelligible fashion and written in standard English?

Reviewer #1: Yes

Reviewer #3: Yes

Reviewer #4: Yes

6. Review Comments to the Author

Reviewer #1: The manuscript by Johannes Freudenberg et al. describes the transcriptomic signature of skin samples from 20 patients with HS. The manuscript is well written, the RNA-Seq is appropriate, and the results and discussion are consistent with the analysis reported. However, there is a strong discrepancy within the entire manuscript (and figures) in reference to “perilesional” and “unlesional” sites used for RNA-seq. Please clarify and modify within the entire manuscript and figures.

It would have been better to compare lesional, perilesional and non-lesional sites but it is still good and satisfying work.

Line 49, please delete “..” before the reference 9.

Lines 52-56: in the context of genetic variations underlying HS susceptibility and severity, it is important to mention first the genes involved in innate and adaptive immunity (specifying it) and then the genes involved in keratinization and autoinflammation pathways. The references are adequate, but I suggest to also include this novel manuscript that highlight the potential causative role of AIM2 in hidradenitis suppurativa pathogenesis (in particular of syndromic HS). The manuscript is: “Moltrasio C, Cagliani R, Sironi M, Clerici M, Pontremoli C, Maronese CA, Tricarico PM, Crovella S, Marzano AV. Autoinflammation in Syndromic Hidradenitis Suppurativa: The Role of AIM2. Vaccines (Basel). 2023 Jan 11;11(1):162. doi: 10.3390/vaccines11010162”.

Genes name must be written in italics. In addition, Cx26 is encoded by GJB2. Please write “GJB2” gene

without “Cx26, GJB2”.

Line 63: Please also consider this recent and interesting work: “de Oliveira ASLE, Bloise G, Moltrasio C, Coelho A, Agrelli A, Moura R, Tricarico PM, Jamain S, Marzano AV, Crovella S, Cavalcanti Brandão LA. Transcriptome Meta-Analysis Confirms the Hidradenitis Suppurativa Pathogenic Triad: Upregulated Inflammation, Altered Epithelial Organization, and Dysregulated Metabolic Signaling. Biomolecules. 2022 Sep 25;12(10):1371. doi: 10.3390/biom12101371”.

Line 67: Please also consider, after “dermcidin” this recent and interesting work: “Tricarico PM, Gratton R, Dos Santos-Silva CA, de Moura RR, Ura B, Sommella E, Campiglia P, Del Vecchio C, Moltrasio C, Berti I, D'Adamo AP, Elsherbini AMA, Staudenmaier L, Chersi K, Boniotto M, Krismer B, Schittek B, Crovella S. A rare loss-of-function genetic mutation suggest a role of dermcidin deficiency in hidradenitis suppurativa pathogenesis. Front Immunol. 2022 Dec 5;13:1060547. doi: 10.3389/fimmu.2022.1060547”.

The quality of the figures is not satisfactory.

Reviewer #3: Excellent work- an important contribution to the field- all concerns and suggestions have been adequately addressed by the authors

Reviewer #4: The authors have clarified several of the questions I raised in my previous review. Unfortunately, some of them have not been fully addressed. There are still some fundamental concerns with the experimental design and with the analysis that must be revised. This means the strong conclusions put forward by this manuscript are not warranted.

(1) The authors aim to perform a meta-analysis using HS RNA-Seq publicly available data, however, the method of choice does not fully support this. Due to a high heterogeneity of samples, controls and methodologies of RNA sequencing and annotation included in this study, a rigorous meta-analysis should be performed to avoid bias. Vote counting does not address samples heterogeneity. It is a relatively simple method of knowledge utilization, whereas meta-analysis requires careful explanation of the discretionary steps taken in analysis. Moreover, the vote counting method is not being clarified in the methods section. The technical details should be well explained to ensure that readers understand exactly what has been done.

Still the group has interesting results, and it would be a pity to not share this information with others that work in the field. Therefore, based on all the points raised, I advise the authors to change the method of analysis. Instead of performing a meta-analysis, the authors could perform a vote counting review of DEGs. This would be an easy solution to address the bias due to heterogeneity, and still find similar results.

7. PLOS authors have the option to publish the peer review history of their article (what does this mean?). If published, this will include your full peer review and any attached files.

Reviewer #1: No

Reviewer #3: No

Reviewer #4: No

---

## [Author Response · Author response to Decision Letter 1]

9 Mar 2023

Reviewer #1: The manuscript by Johannes Freudenberg et al. describes the transcriptomic signature of skin samples from 20 patients with HS. The manuscript is well written, the RNA-Seq is appropriate, and the results and discussion are consistent with the analysis reported. However, there is a strong discrepancy within the entire manuscript (and figures) in reference to “perilesional” and “unlesional” sites used for RNA-seq. Please clarify and modify within the entire manuscript and figures.

It would have been better to compare lesional, perilesional and non-lesional sites but it is still good and satisfying work.

Thank you for pointing this out. This has been modified as non-lesional throughout since all samples described this way were at least 5 cm from a site of inflammation where the lesional biopsy was performed.

Line 49, please delete “..” before the reference 9.

Done.

Lines 52-56: in the context of genetic variations underlying HS susceptibility and severity, it is important to mention first the genes involved in innate and adaptive immunity (specifying it) and then the genes involved in keratinization and autoinflammation pathways. The references are adequate, but I suggest to also include this novel manuscript that highlight the potential causative role of AIM2 in hidradenitis suppurativa pathogenesis (in particular of syndromic HS). The manuscript is: “Moltrasio C, Cagliani R, Sironi M, Clerici M, Pontremoli C, Maronese CA, Tricarico PM, Crovella S, Marzano AV. Autoinflammation in Syndromic Hidradenitis Suppurativa: The Role of AIM2. Vaccines (Basel). 2023 Jan 11;11(1):162. doi: 10.3390/vaccines11010162”.

Thank you for bringing this to our attention as it was published after the most recent draft of our manuscript was completed. We have updated this draft to include it.

Genes name must be written in italics. In addition, Cx26 is encoded by GJB2. Please write “GJB2” gene

without “Cx26, GJB2”.

Thank you for this suggestion. The sentence at line 54 mentioning connexin-26 is structured to say “genes encoding subunits of the gamma-secretase complex, connexin-26…” Since we are not specifically naming a genes in this sentence and describing encoded proteins we would be mixing up proteins and genes if we particularly described GJB2. We may be understanding this comment incorrectly and are happy to adjust this if additional review is requested or during the proofing process.

Line 63: Please also consider this recent and interesting work: “de Oliveira ASLE, Bloise G, Moltrasio C, Coelho A, Agrelli A, Moura R, Tricarico PM, Jamain S, Marzano AV, Crovella S, Cavalcanti Brandão LA. Transcriptome Meta-Analysis Confirms the Hidradenitis Suppurativa Pathogenic Triad: Upregulated Inflammation, Altered Epithelial Organization, and Dysregulated Metabolic Signaling. Biomolecules. 2022 Sep 25;12(10):1371. doi: 10.3390/biom12101371”.

This has now also been cited at line 56.

Line 67: Please also consider, after “dermcidin” this recent and interesting work: “Tricarico PM, Gratton R, Dos Santos-Silva CA, de Moura RR, Ura B, Sommella E, Campiglia P, Del Vecchio C, Moltrasio C, Berti I, D'Adamo AP, Elsherbini AMA, Staudenmaier L, Chersi K, Boniotto M, Krismer B, Schittek B, Crovella S. A rare loss-of-function genetic mutation suggest a role of dermcidin deficiency in hidradenitis suppurativa pathogenesis. Front Immunol. 2022 Dec 5;13:1060547. doi: 10.3389/fimmu.2022.1060547”.

This has now also been cited at line 68.

The quality of the figures is not satisfactory.

Reviewer #3: Excellent work- an important contribution to the field- all concerns and suggestions have been adequately addressed by the authors

Reviewer #4: The authors have clarified several of the questions I raised in my previous review. Unfortunately, some of them have not been fully addressed. There are still some fundamental concerns with the experimental design and with the analysis that must be revised. This means the strong conclusions put forward by this manuscript are not warranted.

(1) The authors aim to perform a meta-analysis using HS RNA-Seq publicly available data, however, the method of choice does not fully support this. Due to a high heterogeneity of samples, controls and methodologies of RNA sequencing and annotation included in this study, a rigorous meta-analysis should be performed to avoid bias. Vote counting does not address samples heterogeneity. It is a relatively simple method of knowledge utilization, whereas meta-analysis requires careful explanation of the discretionary steps taken in analysis. Moreover, the vote counting method is not being clarified in the methods section. The technical details should be well explained to ensure that readers understand exactly what has been done.

We are concerned that a “rigorous meta-analysis” would be more challenging than suggested with this type of data since we are aiming to jointly analyze both microarrays which measure probe intensities on an arbitrary scale and RNA-Seq datasets which count the number of observed transcripts. To establish the right statistical methods to jointly model such divergent data types without introducing new biases is beyond the scope of this manuscript. To nevertheless address the reviewer’s concerns we have decided to remove any reference to the term “meta-analysis” since the method we used is a meta-analysis in broad sense and could be interpreted as misleading. In addition, we have added two sentences to the discussion (line 145) reflecting the reviewer’s point regarding the limitations (line 222) of the vote counting method and we have updated the methods section (line 299) to describe how the HS signature was determined.

Still the group has interesting results, and it would be a pity to not share this information with others that work in the field. Therefore, based on all the points raised, I advise the authors to change the method of analysis. Instead of performing a meta-analysis, the authors could perform a vote counting review of DEGs. This would be an easy solution to address the bias due to heterogeneity, and still find similar results.

We hope this concern has been adequately addressed with changes discussed in the previous comment.

---

## [Decision Letter · Decision Letter 2]

22 Mar 2023

A hidradenitis suppurativa molecular disease signature derived from patient samples by high-throughput RNA sequencing and re-analysis of previously reported transcriptomic data sets

PONE-D-22-15397R2

Dear Dr. Sayed,

We’re pleased to inform you that your manuscript has been judged scientifically suitable for publication and will be formally accepted for publication once it meets all outstanding technical requirements.

Kind regards,

Michel Simon, Ph. D.

Academic Editor

PLOS ONE

Additional Editor Comments (optional):

Reviewers' comments:

Reviewer's Responses to Questions

**Comments to the Author**

1. If the authors have adequately addressed your comments raised in a previous round of review and you feel that this manuscript is now acceptable for publication, you may indicate that here to bypass the “Comments to the Author” section, enter your conflict of interest statement in the “Confidential to Editor” section, and submit your "Accept" recommendation.

Reviewer #4: All comments have been addressed

2. Is the manuscript technically sound, and do the data support the conclusions?

Reviewer #4: (No Response)

3. Has the statistical analysis been performed appropriately and rigorously? 

Reviewer #4: (No Response)

4. Have the authors made all data underlying the findings in their manuscript fully available?

Reviewer #4: (No Response)

5. Is the manuscript presented in an intelligible fashion and written in standard English?

Reviewer #4: (No Response)

6. Review Comments to the Author

Reviewer #4: (No Response)

7. PLOS authors have the option to publish the peer review history of their article (what does this mean?). If published, this will include your full peer review and any attached files.

Reviewer #4: No

---

## [Editor Report · Acceptance letter]

29 Mar 2023

PONE-D-22-15397R2 

A *Hidradenitis Suppurativa* Molecular Disease Signature Derived from Patient Samples by High-Throughput RNA Sequencing and Re-Analysis of Previously Reported Transcriptomic Data Sets  

Dear Dr. Sayed:

I'm pleased to inform you that your manuscript has been deemed suitable for publication in PLOS ONE. Congratulations! Your manuscript is now with our production department. 

Kind regards, 

on behalf of

Dr. Michel Simon 

Academic Editor

PLOS ONE